# Self-Assessment of Cultural Competence and Social Determinants of Health within a First-Year Required Pharmacy Course

**DOI:** 10.3390/pharmacy10010006

**Published:** 2021-12-28

**Authors:** Ulyana Kucherepa, Mary Beth O’Connell

**Affiliations:** 1Clinical Pharmacy Department, SSM Health St. Mary’s Hospital—St. Louis, 6420 Clayton Rd., Richmond Heights, MO 63117, USA; ubkucher@gmail.com; 2Pharmacy Practice Department, Eugene Applebaum College of Pharmacy and Health Sciences, Wayne State University, 259 Mack Ave., Detroit, MI 48201, USA

**Keywords:** cultural competency, cultural competence, cultural competence curriculum, social determinants of health, pharmacy, students, education, assessments

## Abstract

As social determinants of health (SDOH) and health disparities are integrated with cultural competence in healthcare education, tools assessing multiple topics are needed. The Self-Assessment of Perceived Level of Cultural Competence (SAPLCC) survey is validated in student pharmacists and includes SDOH. The research objective was to determine if the SAPLCC survey can quantify cultural competence and SDOH course learning. First-year student pharmacists (N = 87) completed the SAPLCC survey anonymously before and after a social and administrative sciences course. The survey had 75 items with 1–4 Likert scales (4 high, total 300 points). All items were summed for the total score. Each item was assigned to a domain and factor. Factors were assigned to domains. The baseline total score was 190 ± 29 points, increasing by 63 ± 33 points post-course. All domains (i.e., knowledge, skills, attitudes, encounters, abilities, awareness), 13 of 14 factors, and total scores statistically increased. The SAPLCC tool captured student pharmacists’ self-reported changes in cultural competence and SDOH.

## 1. Introduction

Using cultural competencies and resolving suboptimal social determinants of health (SDOH) are considered important to eliminating health disparities globally per the World Health Organization [1] and nationally within countries such as the United States through their Healthy People 2030 public health initiatives [2,3]. “Cultural competence is a set of values, behaviors, attitudes, and practices within a system, organization, and program or among individuals, which enables them to work effectively cross-culturally [4].” Striving to achieve cultural competence is a dynamic, ongoing, developmental process that requires a long-term commitment. This life-long learning to increase one’s cultural knowledge, skills, and applications, known as cultural humility, can improve patient health care outcomes and satisfaction, decrease health disparities [4,5] and should begin in college. Social determinants of health are non-medical factors (e.g., education, income, living environment, working conditions) influencing health outcomes and inequities [6]. Thirty to fifty-five percent of health outcomes are related to SDOH [3,7]. Thus, SDOH also needs to be incorporated into education and practice. 

Therefore, pharmacy education accreditation standards in the United States [8,9] and other countries include cultural competence and SDOH [8,9,10,11] to create culturally competent healthcare professionals who can be actively engaged in providing individualized patient care that includes using the patients’ cultural beliefs, behaviors and expectations; resolving cultural differences; and eliminating health disparities and systemic racism [12]. In the United States and some international pharmacy colleges (e.g., Indonesia, Saudi Arabia), the Accreditation Council for Pharmacy Education (ACPE) standard 3.5 calls for the inclusion of cultural competencies and SDOH in Doctor of Pharmacy curricula [8,13]. The ACPE Appendix I lists cultural awareness as a required element of the didactic curriculum [2]. The United States 2013 Center for the Advancement of Pharmacy Education learning objectives included recognizing cultural identity and norms, having a respectful attitude for others from different cultures, incorporating cultural beliefs and practices into care plans, and assessing health literacy and adjusting communications accordingly [9]. All ACPE-accredited pharmacy programs are required to assess students’ cultural awareness and sensitivity and SDOH knowledge. 

Many cultural competence assessment tools exist but they have limited scope, have not been validated in student pharmacists, do not have rigorous psychometric analyses, do not include SDOH items, and or have not been used for course or curriculum longitudinal assessment [14,15]. Based on limitations of previous cultural competence surveys and the need for multiple surveys, the Clinical Cultural Competency Questionnaire (CCCQ) and California Brief Multicultural Competency Scale (CBMCS) surveys were combined to create the first version of the Self-Assessment of Perceived Level of Cultural Competence (SAPLCC) survey to assess cultural competencies [16]. The CCCQ had been developed to analyze physician cultural competencies and has been validated in student pharmacists. The CBMCS had been developed to analyze mental health providers’ cultural competence. The SAPLCC survey items were divided into six domains and fourteen factors. To further evaluate the new tool, the survey was administered to first and second-year pharmacy and medical students [17]. Factor SAPLCC scores differed by discipline and race but did not differ by gender, language, or international exposure. 

To decrease the number of items and add SDOH items, version two of the SAPLCC survey was created and evaluated in first to fourth-year student pharmacists from eight universities [18]. The survey had good reliability (overall Cronbach’s alpha scores were 0.95 and 0.8 or higher for the factors). The data were further analyzed by program year [19]. Mean scores after the fourth year were below maximum values, suggesting curriculum changes needed and additional education and training required after graduation [16,18]. 

The SAPLCC survey has not been used to evaluate cultural competence development and SDOH learning from a single course. Therefore, this study was designed to see if the SAPLCC survey could measure cultural competencies and SDOH learning in a required pharmacy course and its ability for curriculum longitudinal assessments. 

## 2. Materials and Methods

The participants were first-year student pharmacists in a United States college of pharmacy enrolled in the second social and administrative sciences 16-week course offered in the winter semester. The course contains integrated culture, complementary health approaches, and SDOH components (Appendix A). The culture component consists of 5.75 h of in-class teaching and small group activities, one-hour lecture outside of class time, and three homework assignments. Students in pairs prepared and delivered a complementary health approach presentation. Each student listened to two other complementary health approaches presentations. The SDOH component is two hours of in-class teaching and small group activities. Most classes have some prework to complete to create class time for small group activities. Culturally diverse small groups are created from student-provided cultural characteristics obtained from a questionnaire completed prior to the course. Characteristics included gender, race, ethnicity, religion/spirituality, and country of birth.

Students completed the SAPLCC version two survey [18,19] at the beginning and end of the course. Each survey completion was worth one percent of the final score. Students used an anonymous code assigned by an education specialist not involved with the course to complete the survey on Qualtrics (Provo, UT, USA). Preassigned student codes were used to minimize social desirability bias in responses. This approach helped to maintain confidentiality and avoid privacy issues with the goal to obtain more personal and truthful responses. Investigators did not have access to the code number—student name list. The course coordinator gave the list of codes for completed surveys to the education specialist to determine student names for entering survey completion points in the grade book. This research project received Institutional Review Board exemption via category 4 (protocol number IRB-21-05-3606).

The SAPLCC survey contains five demographic variables described in Table 1. Ethnicity, religion/spirituality, birth country, and current residence were added demographics to the SAPLCC tool. The survey has 75 items described in Table 2. For each item, answer options were not at all (1), a little (2), quite a lot (3), and very (4). Higher scores represent greater cultural competence and SDOH knowledge. The items are divided into six domains—knowledge (16 items), skills (11 items), attitudes (15 items), encounters (11 items), abilities (13 items) and awareness (9 items) [18]. Each domain is divided into two to three factors for a total of 14 factors. Each factor contains two to nine items. The domain, factor, and overall mean scores represent the individual item points for that category divided by the total number of items in that category. The change scores represent the post score minus the pre score. Domain, factor, and total scores were normalized by dividing by the number of survey items in each component. Normalized domain, factor, and total mean scores were categorized into three outcome categories defined as high if 3 or more points, moderate if between 2 and 3 points and low if less than 2 points, cutpoints used in the SAPLCC validation article [19]. 

Descriptive statistics were utilized to obtain mean, median, and standard deviations for survey responses for each factor, domain, and overall survey scores and change data. Pre- and post-course domain, factor, and overall scores were compared using Wilcoxon signed-rank tests. Data were compared across demographic variables using Kruskal Wallis and Mann-Whitney U statistical tests. A p-value of less than or equal to 0.05 was regarded as statistically significant. All analyses were conducted using SPSS version 27 (Armonk, NY, USA). 

## 3. Results

### 3.1. Demographics

The survey response rate was 98% (87 of 89) for first-student pharmacists completing pre- and post-course surveys. Table 1 lists student demographics. The cohort was predominantly female, White, and Christian, however, diversity existed in all demographics. Students’ age ranged from 19 to 45 years old with the majority of students being younger than 25 years old. Thirty-five percent of the students were foreign-born and 5% currently lived in Canada. Forty percent of the students already had a Bachelor’s or Master’s degree.

### 3.2. SAPLCC Domain, Factor, and Total Point Scores

The mean point scores for all six domains, fourteen factors, and total points are represented in Table 2 and Figure 1a,b. At the domain level, the lowest mean point scores prior to the course were observed in skills (19.6 ± 7.1) and encounters (26.1± 7.6). The highest mean point scores in the pre-survey were in attitudes (47.5 ± 7.2) and abilities (36.5 ± 7.9) domains. All domains showed significant increases after the course (*p* < 0.05). The greatest changes in pre and post-course were observed in the knowledge and skills domains. The lowest change was observed in the awareness domain. The total SAPLCC score increased statistically after the course. 

The normalized domain and total mean scores are listed in Table 3. The highest baseline pre-survey mean scores for domains were observed in attitudes and awareness domains (3.2 ± 0.48 and 3.3 ± 0.47, respectively). Pre survey normalized mean scores for encounters and abilities domains were moderate (2.4 ± 0.69 and 2.8 ± 0.61, respectively). Students showed the lowest baseline normalized mean scores in knowledge (1.9 ± 0.51) and skills (1.8 ± 0.64) domains. 

In all domains, post mean scores were in the high category. The maximum mean score of 4 was not achieved for any of the domains. The post total mean score was significantly different than the baseline total mean score.

Changes in mean domain scores were found to be statistically significant in all six domains. The largest changes in mean scores were observed in skills (1.5 ± 0.8) and knowledge domains (1.3 ± 0.6) and lowest in the awareness domain (0.1 ± 0.5). 

### 3.3. Factors

The pre, post, and change normalized factor mean scores are listed in Table 3. All post mean scores were found to be statistically different from pre mean scores except understanding barriers to health care (F13). The highest pre-survey factor scores were in attitudes (F5, F6, F7) and awareness (F12, F13, F14) domains. After the course, five-factor mean scores were 3.5 or higher (F5, F6, F7, F11, and F14).

The amount of self-perceived learning varied among the factors (Table 3). The highest mean score changes were for providing culturally competent services (F3) and dealing with cross-cultural conflicts (F4), both in the skills domain. Five factors increased by 1 or more points, five factors increased by 0.5 to 0.99 points, and four factors increased by less than 0.5 points. 

At the end of the course, the percent of students in the post-high category varied from 56–88% for the domains with attitudes the highest and skills and encounters the lowest. For all factor scores, 61.8% of the students’ individual scores ranked in the high category, 36.8% in the moderate category, and 1.4% in the low category. The number of high category post scores ranged from 45–84% for individual factors. The highest factors were recognizing social determinants of health (F6) and interpersonal/intercultural interactions (F7). The lowest factor value was observed in addressing population health issues (F1). 

Analyzing the post-category percentages should be combined with the amount of self-perceived learning based on normalized mean scores and actual point changes. For example, although factor 1 has the lowest percentage in the post-high category, the amount of change (1.2 ± 0.7 mean change score; 8.6 ± 4.7 point change) is greater than the change in largest percentage in the high category for factor F6 (0.4 ± 0.6 mean change score; 1.9 ± 2.9 total point change) and F7 (0.2 ± 0.6 mean change score; 0.9 ± 2.4 point change).

### 3.4. Demographic Differences

Pre and post-course domain and factor SAPLCC scores were minimally different across the student demographics. Appendix B has the significant p values. Only one domain score was statistically different [religion/spirituality pre-awareness domain] at the factor level, statistically significant differences were observed for 9-factor categories (4 pre and 5 post). Gender differences existed only for pre-managing cross-cultural communication challenges (F9). Factor scores by race were statistically different for pre confronting racial dynamics (F14) and post increasing comfort during cross-cultural encounters (F8). Statistical differences by religion/spirituality group were seen with pre-understanding barriers to health care (F13), pre confronting racial dynamics (F14), post understanding the context of care (F2), post providing culturally competent services (F3), post recognizing SDOH (F6), and post applying multicultural knowledge (F11). 

## 4. Discussion

The SAPLCC tool facilitated students assessing their self-perceived cultural competence and SDOH knowledge and skills related to 75 items prior to the course. Conducting this assessment prior to the course created a formative assessment and self-reflection to the base subsequent course learning. Students began the course with high baseline mean scores in attitudes and awareness domains, intermediate scores in encounters and abilities domains, and lowest scores in knowledge and skills domains. High baseline scores could reflect that most of our students live in diverse communities learning about culture and SDOH from their environments and the fact our college is in a large diverse metro area with many cultures, poverty and unemployment. The minority and foreign-born students in our cohort have learned life experiences influencing their baseline competencies and learning. They also educate other students throughout their lives. 

The SAPLCC tool captured self-perceived learning in all domains and 12 of the 14 factors. Self-perceived learning occurred even in areas with high baseline assessments. Our study supports that first-year student pharmacists can increase their cultural competence and SDOH learning early in the curriculum prior to experiential learning. Increasing cultural competence and understanding of SDOH’s impact on health outcomes and health disparities could impact empathy and practice during future experiential learning activities. Since none of the mean or median post scores reached the maximum point values, this assessment could be used in subsequent years to determine additional cultural competencies and SDOH knowledge and skills learned. 

Our findings correlate with course content that covers 39 of the 75 SAPLCC items and all domains and factors [18] in various degrees of depth and breadth. In Appendix A all course content is linked to factors and domains. Factors with low baseline values and large changes, found in knowledge and skills domains and factors, reflect items with greater coverage in the course. For example, factor 4 provides culturally competent services aligned with learning about the 4 Cs (call, cause, cope, concerns) method of working with patients with potential cultural encounters, medical interpreters, health literacy assessment tools, and bilingual printed or internet patient education materials. The course has no direct patient care, so less learning was understandable in encounters and abilities domains that assess patient care competencies. Course content covered competencies in the awareness and attitudes domains, domains that had the highest baseline scores and lowest change scores. This content might be more reinforcement of these competencies. 

The curriculum committee at our college is using our study information along with curriculum mapping data to identify curricular gaps and opportunities to guide future curriculum changes related to enhancing cultural competence and SDOH education and training in this and other courses. The SAPLCC survey will be used in subsequent years to assess learning across the full four years of our pharmacy curriculum and monitor how cultural competence and SDOH knowledge and skills improve during student progression through the didactic and experiential curriculum. 

The SAPLCC version two was developed and used to assess cultural competence across all four years of pharmacy curricula at a single point in time in student pharmacists from eight universities [19]. When comparing these first-year students to our first-year students’ post-course data, eleven of our fourteen-factor mean scores and the total score (2.6 vs. 3.4, respectively) are greater. Many variables can explain these differences such as curricula content depth, breadth and placement, cultural mix of students in each program, and university location. 

In other studies, evaluating student pharmacists’ cultural competence, attitude changes also did not occur. In one study, a modified Clinical Cultural Competency Questionnaire, which is part of the SAPLCC, was used to assess first through third-year student pharmacists (33% minority students) from eight universities after elective or required cultural competence education [20]. The students had no changes in the attitudes domain but changes in knowledge, skills, and encounter domains occurred. Their hypotheses for why attitudes did not change were previous cultural competence training already created positive attitudes, insufficient activities, minimal time to change attitudes in one course, course activities were not graded that might have decreased student emphasis on learning, and students need to wrestle with topics to result in attitude changes. In another study, second-year student pharmacists had the highest scores in attitudes and lower scores in knowledge, skills, and encounters [21]. In our study, attitudes also had the highest baseline. Our high baseline attitudes domain scores might reflect the cultural diversity of our student pharmacists, faculty, staff, university and metro area. We concur with the previous investigators that insufficient time exists for students to wrestle with the topics to change attitudes. More exposure to courses encompassing culture and SDOH topics in the didactic curriculum, and experiential education to apply learning might allow more opportunity to create changes in attitudes. 

In our study with 24% minority students, race had no impact on domain scores and minimal impact on factor scores (one pre—F14 and one post—F8 were statistically different). In other studies with higher minority student cohorts, greater changes are seen between races. In SAPLCC version one with 78% minority students, race resulted in significant differences for factors 5, 7, 11, 12, and 13 [16]. African American student scores were higher than white student scores for all factors and Asian student scores for all but one factor [16]. In the SAPLCC version two study with 61% minority students from eight universities, race differences were reported for all factors [19]. Thus, minority student population might influence outcomes. The other difference might be our study was just first-year students and the other studies were cohorts of four years of students combined. Using four years of students includes students with more learning opportunities from additional courses and direct patient care. In another study using the CCCQ scale to measure learning after two hours of cultural competence training, differences in learning existed by race and ethnicity also [22].

Both our study and the SAPLCC version two four-year eight university study [19] evaluated impact of gender and age on SAPLCC scores. In our study, gender had almost no impact (only pre F9 significantly different). In the eight university study, gender influenced four of the fourteen-factor scores (F2, F5, F7, and F9) with female students higher for factors 5 and 7 and male students higher for factors 2 and 9. In terms of age, no impact was seen in our study. In the four-year student cohort study, six-factor scores (F1, F3, F4, F7, F12, F14), were significantly different with older students doing better on F1, F3, and F4 and younger students doing better on F7, F12, and F14. Gender and age appear to have no to minimal impact on cultural competence and SDOH self-perceived learning. Study sample sizes might be too small to identify true differences, however, if differences existed most likely they would be small. 

Since many health care decisions reflect cultural beliefs, we included students’ religion/spirituality backgrounds in our assessments. Religion influenced differences in two pre-awareness domain factors (F13, F14) and four post factors, one in knowledge (F2), skills (F3), attitudes (F6), and abilities (F11) domains. These differences could be explained by multiple comparisons without corrections or differences in lived life and health care experiences personally or for their families prior to pharmacy school. Unfortunately, the SAPLCC tool has no questions related to health care decisions, experiences, or comfort-related to carrying for patients with different religions and spirituality. We plan to add these questions for subsequent SAPLCC use for our curriculum assessments.

A parameter we did not assess was previous cultural competence training. In a study designed to look at emotional intelligence and cultural competence, first-year student pharmacists who had previous training had higher self-cultural awareness [23]. Their cohort had 71% Black student pharmacists. Interestingly they found higher self-cultural awareness was not associated with higher cultural competence on the Quality and Culture Quiz, a cultural competence measurement. The impact of prior cultural competence training on cultural competence was also confirmed in second-year student pharmacists [21]. The percent of students with prior cultural competence training can be high and can increase over time [22]. In 2010, 27% of the third-year students had prior training, which increased to 63% in 2011 and 57% in 2012. External training at internship sites, missionary work, and military experience accounted for previous training. This study also documented significant differences in some cultural competence factors between those students with and without prior cultural competence training. We will add this demographic for subsequent use of SAPLCC.

None of our post normalized mean scores reached the maximum points for factor, domain or total scores. Thus, we could use this survey for assessments later in the program. Additional learning over the next 3 years could help reach full competence for all measurements for all students. Cultural competence is a life-long journey so even if a student reaches maximum points, dedication post-graduation to enhance cultural knowledge and skills is important (cultural humility). 

Evidence exists that this learning does occur across a pharmacy curriculum. With both versions of SAPLCC, factor, domain and total scores increased with time [16,19]. Of note, some scores dropped in the third-year students and then increased in the fourth year [16]. With the SAPLCC version two study, mean scores were usually higher in the fourth year with third- and fourth-year scores being similar [19]. Other studies using different tools, also find student pharmacists continue to learn but have not received the maximum points on their assessment scales [22,23,24,25]. 

The need to include more didactic and experiential education to develop cultural competence and identify and resolve SDOH problems have been identified in other studies and reports [21,26,27]. A recent survey of American and Canadian colleges of pharmacy curricula found significant variability and completeness of cultural competence, SDOH, and health literacy taught over the four years of pharmacy curricula [5]. In keeping curricula responsive to health care and societal needs, a call for further training incorporating antiracism from personal and structural aspects into cultural competence and SDOH training has been published [27]. Best practices for teaching cultural competence, SDOH, health disparities, and social justice are not known. Using tools such as the SAPLCC can help identify culture and SDOH curriculum gaps, the latter not a major component of other cultural competence tools, and areas for future education and training enhancements. 

Since cultural competence is a lifelong journey extending beyond reaching maximum scores and formal education, the abilities of educators and preceptors to further develop the students and health care professionals is a critical component [5,21,22]. College and health care practice educators and preceptors also need to develop their academic skills to create better student and health care professionals to be culturally competent, who can then help decrease health disparities as our world continues to become more diverse [5,21,26]. Educators and practitioners need to model the skills, attitudes, empathy and compassion taught in their courses and at their practice sites to facilitate student pharmacist growth and development in these areas [5]. Professor and provider culturally incompetent behaviors and actions can occur. One college of pharmacy with a diverse student population evaluated student inclusion and respect within their program and found black students having more negative experiences than other minority or white students, some of which related to educators and preceptors [28]. In the future to advance student cultural competence, educator and practitioner cultural competence needs to be advanced and assessed as well. 

Our study has some strengths with using a validated tool in student pharmacists, having a high response rate, and using anonymous nature of survey completion. Our study had some limitations, some of which were reported in other studies. Our study, like some others [21,22], were single-center studies designed to analyze a college-specific course or program. Some studies used multiple universities however they combined students with different cultural and SDOH curricula into single categories [19,24]. Both study types have value. Although our study had students complete the survey anonymously to facilitate honesty in responses, social desirability bias could still exist. This concern was expressed in other SAPLCC studies and with other cultural competence tools. We did not assess for previous cultural competence education and training, which could influence student baseline values and result in different amounts of self-perceived learning. The SAPLCC tools as well as most other cultural competence assessment tools rely on self-perceived assessments with the possibility of higher initial self-assessments. The impact is that the reported amount of learning can be less since the tool is based on falsely high pre-self-assessments. After learning and experiential education, students better understand their competencies and more realistically evaluate post-learning, thus differences between post and baseline might not reflect total learning. The potential for overconfidence with self-assessments has been discussed in other cultural competence studies. Actual evaluations on rotations or using culture-based patient simulations would be better assessments of cultural competence. Longitudinal assessments can help evaluate the sustainability of learning. 

Controversy exists with analyzing survey data with parametric tests. The SAPLCC studies [16,17,18,19] used parametric testing. One of their justifications is transforming ordinal item data to a combined domain, factor, and total score creates a continuous variable. Most but not all domain, factor, and total point score data in our study had reasonably normal distributions. However, when we assessed normalcy for the normalized domain and factor scores, most were not normally distributed. Thus, we analyzed our data with nonparametric tests. None of the cultural competence studies reviewed corrected for multiple comparisons treating each domain, factor, and total score as a separate concept. Thus, some of our significant findings might be related to chance, especially for the comparisons by the various demographics. 

## 5. Conclusions

The SAPLCC survey captured first-year student pharmacists’ cultural competence and SDOH self-perceived learning within a social and administrative course. All post mean normalized scores were less than the scores’ maximum points, suggesting further didactic and experiential education and training to further enhance student pharmacists’ cultural competence and SDOH knowledge. No ceiling effect was seen so this tool can be used in subsequent years of the pharmacy curriculum. Overall student demographics had little effect on SAPLCC scores. This tool could be used at other pharmacy colleges, with results potentially different depending on placement in programs and the extent of coverage of the various topics across curricula.

## Figures and Tables

**Figure 1 pharmacy-10-00006-f001:**
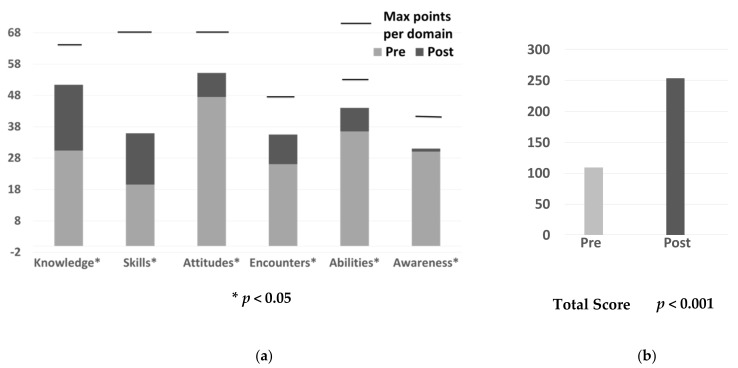
Pre- and post-survey domain and total mean point scores. (**a**) Pre- and post-domain mean point scores; (**b**) Pre and post total point scores.

**Table 1 pharmacy-10-00006-t001:** Cohort demographics.

Category	Number, (%)
Gender Female Male	58 (66.7) 29 (33.3)
Race White Asian Black/African American	66 (75.9) 18 (20.7) 3 (3.4)
Ethnicity Arab American Hispanic/Latinx Neither	32 (36.8) 1 (1.1) 54 (62.1)
Age (years) ≤24 25–29 ≥30	70 (83.3) 8 (9.5) 6 (7.1)
Education prior to pharmacy school High school diploma Associate degree Bachelor’s degree Master’s degree	38 (43.7) 14 (16.1) 33 (37.9) 2 (2.3)
Religion/spirituality Christianity Islam Agnosticism Atheism Buddhism Hinduism Sikhism Spiritual	46 (53.5) 29 (33.7) 4 (4.7) 3 (3.5) 1 (1.2) 1 (1.2) 1 (1.1) 1 (1.1)
Birth country and current residence Born United States, live in Michigan Born Canada, live in Michigan Born Canada, live in Canada Born another country, live in Michigan Born another country, live in Canada	57 (65.1) 2 (2.3) 3 (3.5) 24 (27.9) 1 (1.2)

**Table 2 pharmacy-10-00006-t002:** SAPLCC survey description and domain, factor, and total score points before and changes after course.

Domains and Factors	Number of Survey Items	Maximum Points	Factor Titles	Pre Point Score Mean ± SD	Percent of Maximum Points at Baseline	Points Change ^1^ After Course Mean ± SD
Knowledge domain	16	64		30.4 ± 8.2	47.5	21.0 ± 9.5
Factor 1	7	28	Addressing Population Health Issues	13.7 ± 3.9	48.9	8.6 ± 4.7
Factor 2	9	36	Understanding the Context of Care	16.7 ± 4.8	46.4	12.4 ±6.0
Skills domain	11	44		19.6 ± 7.1	44.5	16.3 ± 8.6
Factor 3	7	28	Providing Culturally Competent Services	12.3 ± 4.5	43.9	10.3 ± 5.6
Factor 4	4	16	Dealing with Cross-Cultural Conflicts	7.2 ± 2.8	45.0	6.0 ± 3.4
Abilities domain	15	60		47.5 ± 7.2	79.2	7.7 ± 7.1
Factor 5	6	24	Recognizing Disparities-Related Discrimination	18.6 ± 4.4	77.5	3.1 ± 4.2
Factor 6	5	20	Recognizing Social Determinants of Health Improving	16.8 ± 2.7	84.0	1.9 ± 2.9
Factor 7	4	16	Interpersonal/Intercultural Interactions	14.0 ± 2.7	87.5	0.9 ± 2.4
Encounters domain	11	44		26.1 ± 7.6	59.3	9.4 ± 8.3
Factor 8	3	12	Increasing Comfort During Cross-Cultural Encounters	8.5 ± 2.3	70.8	1.6 ± 2.3
Factor 9	8	32	Managing Cross-Cultural Communication Challenges	17.5 ± 5.9	54.7	7.8 ± 6.7
Abilities domain	13	52		36.5 ± 7.9	70.2	7.5 ± 7.4
Factor 10	8	32	Assessing Population Health Needs	22.4 ± 5.5	70.0	4.3 ± 5.1
Factor 11	5	20	Applying Multicultural Knowledge	14.1 ± 3.1	70.5	3.3 ± 3.3
Awareness domain	9	36		30.1 ± 4.3	83.6	0.9 ± 4.5
Factor 12	3	12	Engaging in Self-Reflection	10.6 ± 1.6	88.3	−0.6 ± 2.4
Factor 13	4	16	Understanding Barriers to Health Care	12.8 ± 2.3	80.0	0.0 ± 2.6
Factor 14	2	8	Confronting Racial Dynamics	6.8 ± 1.4	85.0	0.4 ± 1.4
Total score	75	300		190.2 ± 29.3	63.4	62.9 ± 32.9

^1^ Amount of learning calculated as total points after course minus total baseline points for each domain, factor, and total score.

**Table 3 pharmacy-10-00006-t003:** Pre and post course normalized SAPLCC mean and median scores.

Domains and Factors (Number of Items)	Normalized Score ^1^ Mean ± SD		Mean Change Scores ^1^ Mean ± SD	Median Score ^2^		Percent Students ^3^			Percent Students ^3^		
						Low Score <2	Mod Score 2–3	High Score >3	Low Score <2	Mod Score 2–3	High Score >3
	Pre	Post		Pre	Post	Pre			Post		
Knowledge domain Factor 1 (7) Factor 2 (9)	1.9 ± 0.51	3.2 ± 0.47	1.3 ± 0.6	1.9	3.2 *	63%	34%	3%	1%	42%	57%
	2.0 ± 0.56	3.2 ± 0.58	1.2 ± 0.7	2.0	3.0 *	39%	58%	3%	1%	54%	45%
	1.9 ± 0.54	3.2 ± 0.48	1.4 ± 0.7	1.8	3.2 *	63%	34%	3%	1%	40%	59%
Skills domain Factor 3 (7) Factor 4 (4)	1.8 ± 0.64	3.6 ± 0.59	1.5 ± 0.8	1.7	3.3 *	63%	32%	5%	1%	43%	56%
	1.8 ± 0.64	3.2 ± 0.59	1.5 ± 0.8	1.6	3.3 *	66%	29%	5%	1%	45%	54%
	1.8 ± 0.71	3.3 ± 0.65	1.5 ± 0.9	1.8	3.3 *	51%	46%	3%	1%	47%	52%
Attitudes domain Factor 5 (6) Factor 6 (5) Factor 7 (4)	3.2 ± 0.48	3.7 ± 0.39	0.5 ± 0.5	3.1	3.9 *	1%	39%	60%	1%	11%	88%
	3.1 ± 0.74	3.6 ± 0.48	0.5 ± 0.7	3.0	3.8 *	6%	50%	44%	1%	23%	76%
	3.4 ± 0.55	3.7 ± 0.39	0.4 ± 0.6	3.4	4.0 *	3%	34%	63%	1%	15%	84%
	3.5 ± 0.67	3.7 ± 0.49	0.2 ± 0.6	4.0	4.0 *	1%	34%	65%	1%	16%	83%
Encounters domain Factor 8 (3) Factor 9 (8)	2.4 ± 0.69	3.2 ± 0.59	0.9 ± 0.8	2.2	3.2 *	29%	55%	16%	1%	43%	56%
	2.8 ± 0.76	3.4 ± 0.56	0.5 ± 0.8	3.0	3.3 *	13%	57%	30%	2%	43%	55%
	2.2 ± 0.74	3.2 ± 0.64	1.0 ± 0.8	2.0	3.1 *	40%	49%	11%	3%	46%	51%
Abilities domain Factor 10 (8) Factor 11 (5)	2.8 ± 0.61	3.4 ± 0.43	0.6 ± 0.6	2.8	3.3 *	6%	68%	26%	1%	35%	64%
	2.8 ± 0.68	3.3 ± 0.47	0.5 ± 0.6	2.9	3.1 *	5%	73%	22%	1%	47%	52%
	2.8 ± 0.61	3.5 ± 0.47	0.7 ± 0.7	2.8	3.4 *	5%	74%	21%	1%	40%	59%
Awareness domain Factor 12 (3) Factor 13 (4) Factor 14 (2)	3.3 ± 0.47	3.5 ± 0.37	0.1 ± 0.5	3.3	3.6 *	1%	25%	74%	1%	16%	83%
	3.5 ± 0.54	3.3 ± 0.56	-0.2 ± 0.8	3.7	3.3 *	2%	35%	63%	3%	36%	61%
	3.2 ± 0.58	3.2 ± 0.39	0.7 ± 0.7	3.0	3.5	1%	55%	44%	1%	36%	63%
	3.4 ± 0.71	3.6 ± 0.56	0.2 ± 0.7	3.5	4.0 *	2%	37%	61%	1%	27%	72%
Total score (75)	2.5 ± 0.39	3.4 ± 0.36	0.8 ± 0.4	.2.5	3.4 *	5%	87%	8%	1%	17%	82%

SD = standard deviation, Mod = moderate;^1^ Range is 1 to 4 with 4 high; Sum of item scores per domain, factor, or total score were divided by number of items in domain, factor, or total score to create normalized scores for comparisons to each other since different points in the various domain, factor, and total scores; ^2^
*p* values * equals significantly different. All *p* -values ≤ 0.001 except for awareness domain (*p* = 0.031), F12 (*p* = 0.016), F13 (*p* = 0.607), and F14 (*p* = 0.007); ^3^ Number of students (N = 87) whose individual normalized domain, factor, and total scores fell into low, moderate and high categories.

## Data Availability

The data presented in this study are available on request from the corresponding author. The data are not publicly available due to unfunded research and lack of student permission to share.

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
