# Peer review of "Self-Assessment of Cultural Competence and Social Determinants of Health within a First-Year Required Pharmacy Course"

_pharmacy, 2021, doi:10.3390/pharmacy10010006_

Round 1

Reviewer 1 Report

I appreciate the opportunity to review this important work. I think this study adds to the growing body of literature to help determine best practices to create a culturally responsive pharmacist workforce and would be of interest to faculty of schools and college of pharmacies.

Though it looks like there are a number of items to address, most of these are copy edits and points needed for clarity. There is one item related to a statistical test that may need follow-up/verification (mean versus median on the Likert scale data in table 4). Overall, the study appears to be designed well, and the authors did a great job of comparing results to what is already known. Some minor additional details are suggested in the methods as described below. I would like to see more specific discussion about how the curriculum content connects to their results (ie do any of the results make sense for what is included or not in the course and do they think the course needs modification based on what they saw and if so how and what does that mean to readers).  Please see specifics below in the table.

I am attaching this as a document as well in case the table doesn't come through well.

Manuscript Location

Reviewer’s Comment

Title and throughout (copy edit)

Consider being consistent with terms competency vs competence throughout manuscript

Throughout (copy edit)

Please verify, but I think the reference numbers should be located before the punctuation versus after for this journal.

General

Consider a crosswalk of your course content to the knowledge domains and factors (perhaps in the tables or an additional supplementary document if there is not room). This could also help you interpret the results and explain how you may consider course modification (see comment in discussion about this).

Abstract Line 19

Consider removing demographic results from abstract

Abstract

What do the course factor and domain scores means?

Abstract Line 20 (copy edit)

Should this read “All domains” versus “all domain”

Abstract Lines 20 and 21

Please specify that the scores improved vs domains and factors improving

Abstract lines 21-22

The following sentence is not clear to me as the reader. “Post course factor and domain means were 87% above 3, 11% between 2 and 3, and 2% less than 2.” Consider something like…. “Eighty seven percent of post course factor and domain means were above 3….”. Please clarify.  I could not find these results restated in the manuscript body. Are they in the table. Also, consider removing this as part of the abstract unless you can describe what the three categories mean in the abstract as presently reader may confuse with the likert scale numbers.

Introduction (general)

Since paper focuses on assessing cultural competence, consider introducing the importance of that sooner in the introduction.

Introduction Line 30

The intro beings with “many definitions of cultural competency exist”, and the next line gives only one definition. How did you arrive at the one you chose? Consider restricting those first two sentences. Could say something like, “Cultural competence is frequently (or some other word) as….”  Or describe common elements of the definition to come up with your own,

Introductions Lines 32-35 (copy edit)

“Striving to….” Sentence is long. Commitment is in the sentence twice.  Consider breaking up after applications.

Introduction line 39 (copy edit)

Should this read “Healthy People 2030” instead?

Introduction

Consider adding a transition statement between first and second paragraphs to better connect how social determinants of health connect to cultural competence

Introduction Line 51

You discuss US accreditation standards related to cultural competence and SDOH. Are there any similar standards for education in pharmacy curricula in other countries to mention for your global readers? If not, consider clarifying that there are not.

Introduction line 53 (copy edit)

Should this be “cultural identity” versus “culture identify”

Introduction line 56 (copy edit)

Should this read as a “required element” versus “as a required elements?”

Introduction line 81 (copy edit)

Should “Overal” be “Overall”

Introduction line 84 (possible copy edit) and discussion line 270)

Consider use of curricula versus curriculums. Not sure which is preferred for a global audience.

Introduction line 85 (copy edit)

The sentence starting with “the SAPLCC” is difficult for me to understand as a reader. As written, it sounds like SAPLCC has not been use to evaluate the cultural competence of a course, when I wonder you are referring to evaluating the impact of a course on students’ cultural competence?

Introduction

You captured the literature gap and described your purpose well.

Methods

Suggest clarifying to a global audience that your participants are in the USA

Methods

Suggest clarifying the duration (# weeks) of the course

Methods line 102 (copy edit)

Should this read  ‘in Qualtrics” versus “on Qualtrics”

Methods line 102 (copy edit)

Consider breaking up the long sentence “Preassigned…”

Methods lines 110 and 116 (copy edit)

Consider being consistent with how SAPLCC version two is written (2 or two). If it is formally called SAPLCC 2 could use that. Otherwise, consider version two.

Methods line 122 (copy edit)

The way the factor division is written is not clear to me as a reader. Consider changing to …”Each domain is divided into two or three factors.”

Methods

From prior studies, are there validated norms/definitions of high or low total scores or was this something the investigators decided upon?

Table 1

The factor titles are not always aligned clearly with the domain and factor names.

Methods

Consider adding which demographic variables you collected and why.

Methods line 130

Please clarify what “1 outside of class lecture” specifies in the text (1 hour or one lecture). In the table it looks like this is one additional hour.

Methods line 133

Consider clarifying how you created culturally diverse groups (and what characteristics you used to accomplish that).

Table 2

I like how you specified the cultural and SODH components of the course for the reader.

Table 2

Would begin each point or statement the same (ie with a verb or noun). Presently they vary (ie. “Health literacy” vs Watch)

Table 2

For sections that do not have assigned homework, would clarify as “none” and indicate “0” for those sections that do not have course percent points

Table 3

Did you ask for nonbinary gender? Did you inquire about other races than the ones listed? Did students have the option to select “none” for religion/spirituality?

Figure 1

Consider adding the statistical significance to the pre post total score graph, similar to how you did with each domain

Results line 157

The phrase “the most learning was observed” feels awkward to me as a reader and also gives meaning to the results that you want to describe in the discussion instead. Perhaps “the greatest change”?

Results (Factors) and Table 4

I am not a statistical expert, so please check me on this. From what I recall, there is controversy on using average or mean for Likert scale items because there is no exact difference between each of the measurements, making the data ordinal. This suggests you should use median instead of mean. For your other analyses where you summed the total scores for pre post change and there is a prescribed difference in lower versus higher scores, I think it is probably ok that you used means to look at pre post scores.

Results (Factors) Line 191

Consider not ascribing meaning (ie learning) to change in scores until you get to the discussion. Consider using increase in score or something similar instead.

Results Line 197

The following sentence, “Student post mean domain, factor, and total scores were categorized into three
outcome categories defined as high if 3 or more points, moderate if between 2 and 3 points
and low if less than 2 points.”, seems like it should go into the methods. Please also describe how you arrived at that determination (previously done that way in the literature, etc.).

Results line 205

Please see prior comment about stating “amount of learning” in the results

Discussion

Appreciated this summary paragraph

Discussion

Excellent work connecting your results to the previous literature

Discussion lines 275-386

You describe why authors postulated attitudes did not change in the CCC study. Consider adding your thoughts on why attitudes did not change in your study.

Discussion line 300

Consider omitting pharmacotherapy from the sentence or using another word.

Discussion lines 302-305 (copy edit)

The following sentence… “Of note, our study” is long and I had difficulty understanding it. Dividing it could help with clarity. Also, I was confused by the phrase “assessing evaluated race”

Discussion line 317 (copy edit)

Consider adding a comma after F4

Discussion

Line 359. Could a reason for the lack of maximum impact also be awareness from the participants that one never arrives with cultural competence [that  you acknowledge (cultural humility)]?

Discussion line 377  (copy edit)

Should this read “latter” versus “later”

Discussion

Consider adding some information about how you will use these results to specifically improve your curriculum and what that could mean to your readers with how they will use the information moving forward (things they need to be mindful of when creating or modifying this type of course of curriculum) [see prior comment about how you could look at the domains and factors and where your current course content covers these factors].

Discussion-limitations

Great work with limitations. You identify that a limitation is that you used a self-perception tool. Consider adding that future research should include other measures of knowledge change/learning.

Discussion- limitations

Would also add that another limitation is that your assessment is immediate post and that you do not know that change was sustained. You could mention that longitudinal assessments in future studies could help determine the impact of knowledge change over time.

Conclusions

You describe how the data is being used at your institution. What are the implications for other institutions outside of your college?

Author Response

Thank you for your time in reviewing our manuscript. We highly value your comments and suggestions. Please, see attached file in response to your comments. 

Reviewer 2 Report

Thank you for the opportunity to review this manuscript. Please consider my comments as constructive in nature.

Overall, I find this study to be of some interest to the academic community. It is reasonably well designed, the methodology is sound, and it has good citation and support throughout. I do feel the length of this manuscript is extensive and requires considerable revision to provide a more concise paper and focus the findings better. There is so much verbiage that it is easy for the reader to get lost.

Abstract

  • The content and flow of the abstract is mixed. The initial statements setting up the study are great, but the description of the survey, results, and conclusions are not clearly conveyed. In particular, the inclusion of demographics here is unnecessary. The conclusion statement that the tool captured improved cultural competency and SDOH learning is not adequately supported from the described results in my opinion.
  • There are typos in this and other sections that should be rectified.

Introduction

  • This section is quite lengthy and would benefit from revision to provide a more concise background.
  • The lead-off in this section is a bit confusing. It states that many cultural competency definitions exist, then provide one definition in quotes that is not supported with a citation. The 3rd sentence is so long that I have to re-read it several times before it makes sense. It comes across as if you’re trying to blurt out as much information as possible and it is difficult to follow.
  • There is an excessive quote included in the 2nd paragraph that should be considered for removal. The intent can be revised and included, but I think inclusion of the quote itself is detrimental.
  • In paragraph 3, there’s a statement about examples of learning objectives, but it is unclear whose they are, where they came from, or how they apply. The whole paragraph seems to jump all over the place when reading.
  • I appreciate the in-depth discussion of the survey tools and the subsequent creation of the SAPLCC, however is the complete explanation fully necessary, or could the development and validation be more succinct?
  • Last paragraph is a great lead-in to the value and importance of this study.

Methods

  • There are two statements in this section that are better suited in the results: how many participants there were in the class and what the median value was for responses. This can be revised.
  • The SAPLCC is described and abbreviated again in the 2nd Can be revised.
  • The 3rd paragraph has a rationale for switching from previous versions of the SAPLCC with citations, but I feel that this is unnecessary for two reasons. First, the study is evaluating only this one class, so the rationale for switching is not applicable in this capacity. Second, there’s supporting evidence for the switch (which is fine), but it’s not really needed. It’s great that there’s validation to the survey as previously described, but that doesn’t necessarily need to be discussed here as well as the introduction. A simple “This survey has been validated in student pharmacists” statement with citation would be sufficient from my perspective.
  • I genuinely appreciate and support the breakdown of the SAPLCC v2 survey, but the results data from Table 1 should be removed from the methods section and addressed in the results.
  • The choice of parametric test analysis vs nonparametric is questionable. Though alluded to in the limitations section and a reasonable explanation is provided as to the indication, if nonparametric tests were utilized, this would not need to be addressed. Analyzing nonparametric data with parametric tests can result in incorrect statistically significant findings, while analyzing parametric data with nonparametric tests does not have this concern. Consider reporting your data with nonparametric analysis.

Results

  • This section is quite wordy and would benefit from revision. With the provided tables and graph, the data are more clearly seen visually. Readers should get a summary of the findings and referred to these visuals. Each of 3.2 through 3.4 can be considerably reduced in content.
  • The definitions of low/moderate/high should be moved to the methods section with the survey description. As it’s written, it’s defined twice in the results section (3.2 & 3.3).
  • In 3.3, there’s a statement about combining post-average scores with the amount of learning based on normalized average score, but the reasons for it aren’t discussed. It could be argued that the learning cannot be measured here but the perception of learning could be. Consider adding rationale to the discussion section for this statement.

Discussion

  • This section is incredibly long and needs to be reduced. A lot of the arguments in the section are valid and utilize appropriate support, but need refinement in how they are currently described.
  • Some of the statements on learning I think are not quite justified with the data. The SAPLCC was validated in the self-assessment of cultural competency and SDOH, but the learning component is not definitively captured with this tool. A comparison of student performance in the classroom to the responses on the survey would be a more effective method of analyzing learning. The statement about the tool capturing additional learning is likewise debatable.
  • Reiteration of some of the findings in the 1st paragraph are unnecessary. Consider removing.
  • The second paragraph talks about the different versions of the SAPLCC, but only v2 was used in this study, so why does there need to be such discussion on the previous version?
  • The statement on differences between the validation study including 8 universities and this study attributed to curricula is reaching. I don’t feel that this can be definitively stated as it is right now. There are far too many variables that have not been analyzed to make such a conclusive statement.
  • The 3rd paragraph has a nice summary of two studies that looked at attitudes, but there’s no connection to this study. Consider revising.
  • Despite the subanalysis involving race, gender, and religion yielding limited findings, it makes up half of the discussion section. It feels like you’re trying to make connections that cannot be wholly supported by the size and nature of this study, especially when it’s a one-year study compared to two four-year studies. Consider revising.
  • A self-described weakness of the study is the omission of previous cultural competency training. Literature is reviewed and highlights how this can impact findings, but nothing is said about how it could potentially (negatively) influence these findings. This should be addressed.
  • There is some discussion about post-mean scores not reaching maximum points for a particular factor and suggesting additional learning is required, but this isn’t entirely accurate. While it’s true that additional learning is required, it could be argued that even with achieving max points for a factor additional learning would still be required. This statement assumes that the students perceived a level of achievement or proficiency that isn’t necessarily possible. Consider revising.

Conclusion

  • Once the other sections are revised, I suggest rewriting the conclusion section. It does not adequately summarize the study and its findings.
  • Don’t need to redefine SAPLCC.
  • The 2nd statement about demonstration of learning with room for further growth I don’t think can be made as is. The scores showed improvement in self-perceptions, but not necessarily learning.
  • The statement on curriculum committee use does not belong in the conclusions section. If anything, it should be in discussion as a future direction.

Tables/Appendix

  • Table 1 is rather confusing. It seems to convey two separate ideals – one is on domains and factors, and the other is on performance of each factor. The data just aren’t clearly interpretable from my perspective and could be better suited to separate the data. Where it’s included in the middle of the document (Methods) is further confusing to me because it contains results data. Consider revising or breaking it down into two tables.
  • Table 2 is an excellent breakdown of course content and work, though I think it could be beneficial as an Appendix vs included as a table. This content is not discussed in any other area of the manuscript and is I think more valued as a supplement rather than included as a table specifically.
  • Appendix A does not hold any particular value to me as a reader. The table doesn’t tell me anything other than statistical significance, and even these were limited. Consider removing.

Reviewer 3 Report

Comments:

This is an interesting study, with important insight and implications for not only how we measure cultural competency, but also for pharmacy curriculums. I believe it would be interesting to an international audience. 

  • Suggest streamline introduction for clarity. Paragraph 2 (lines 42-48) is unclear, and could benefit from removal of lengthy quote (line 43-47), and explanation of Many Health People 2030 explanation (may not be familiar to all readers).
  • Suggest move Table 1 and it’s discussion to results.
  • Suggest include explanation of cultural competency and SOD curriculum within program and within the course, prior to discussing the pre-post survey results (e.g. retain in methods prior to introducing survey).
